# "We Need a Father and a Mother!" Rationalities around Filiation in the State: The Invisibility of LGBTIQ+ Families

Rodolfo Morrison [ID], Gabriela Moreno Yates *[ID], Jessica Hormazábal Quiroz, Francisca Galdames Baumann and Pablo Olivares-Araya

Departamento de Terapia Ocupacional y Ciencia de la Ocupación, Facultad de Medicina, Universidad de Chile, Santiago 8320000, Chile
* Correspondence: gmorenoyates@gmail.com

**Abstract:** Objective: This article has two objectives. The first is to describe the procedures, characteristics, and, above all, the rationalities present in three Chilean State institutions in matters of filiation. The second is to analyze how these rationalities impact families that are not represented in public policies, such as LGBTIQ+ families. Method: A documentary analysis was used. The analysis focused on official documents, freely accessible, from three public institutions, understood as local centers of experience. Specifically: (a) the Assisted Reproduction Program of the National Health Fund (FONASA); (b) the State Adoption Office "Mejor Niñez" [Better Childhood]; and (c) the Civil Registry. For the above, approaches to governmentality and post-structuralist analysis of public policies within a documentary analysis methodology were considered to be theoretical–conceptual supports. Results and analysis: The findings reveal a general lack of mention of LGBTIQ+ families and a heteronormative structure in the process of designing official documents from the State. This may exclude these families from public policies. Conclusions: It is concluded that a broader and more diverse understanding of the problems that the State should seek to represent would contribute to a greater representation of diversity in public policies.

**Keywords:** adoption; Chile; governmentality; heteronormativity; kinship; LGBT studies; public policies



## 1. Introduction

In Chile, although there are studies that have addressed the experience of parenthood and family formation among LGBTIQ+ people [1] [1–7], there are still few studies on LGBTIQ+ families and their relationship with the political actions of the State. Some of these investigations have focused on the study of LGBTIQ+ people from two perspectives: (a) one from a human rights point of view [8–10]; and (b) from a post-structuralist analysis of public policies [11–14], highlighting those related to the civil union agreement [15–17]. In this context, this research is positioned from the second group of investigations, seeking two objectives: (a) to describe the procedures, characteristics, and, above all, the rationalities present in three institutions of the State of Chile in matters of filiation; and (b) to analyze how these rationalities, positioned in heteronormativity, exclude families that are not represented in public policies, such as LGBTIQ+ families.

This research represents an effort to generate knowledge about the relationship between the State of Chile and LGBTIQ+ families. The documents incorporated the provision of an approximation to these links, contributing to the initial understanding of the State's representation of the problem and its materialization in the main public policies on filiation, which could include or exclude (through omission or explicitness) LGBTIQ+ families within their actions.

The documentary description of the link between the actions of the State and the filiation of LGBTIQ+ families is considered a basic milestone for critical analysis and the construction of knowledge that guides both the training of professionals and officials at the

State level, as well as the visualization of the latent possibilities of action by the State. In addition, this research arises from a process of consultation with different Chilean LGBTIQ+ families, highlighting the value that the subject of study has for the population involved and the existence of a gap in the available knowledge about the filiation of LGBTIQ+ families in the different territories of Chile.

### 1.1. Problematization

In Chile, in 2021, the Equal Marriage law was approved [18], allowing same-sex couples to marry, adopt, and facilitate the recognition of children. This was achieved thanks to different groups of LGBTIQ+ people who have demanded that the State recognize their ties as a family and their relationships of kinship and filiation [19], as one more of the multiple demands emanating from the historical abandonment of the State regarding the rights of sexual diversity [2] [13,19–23].

It is a fact that LGBTIQ+ families constitute a transgression of the traditional ideal of the family [24], since they can be understood as subjects that establish affective relationships between non-heterosexual or cisgender people and that constitute agreements on kinship and the upbringing of children outside the margins installed by the heterosexual regime [2,4,5,25–29]. Thus, these subjects can establish monogamous or non-monogamous bonds [24,30], seek various reproduction strategies (sexual, assisted, surrogate, adoption, or others) [25], decide not to ascribe to legal ties or refuse to live together or share economic expenses or other aspects of the conservative traditional families logic [31]. In addition, single parents are also considered to be families, i.e., where there is only one LGBTIQ+ person in charge of children (or in the process of gestation or adoption) [32].

In this context, some questions that guide this research are: do State programs in terms of filiation include LGBTIQ+ families? How are these families represented, characterized, or understood in these programs? And more specifically: what do official State documents state when LGBTIQ+ families decide and seek to have children by resorting to the public system, either by assisted reproduction or adoption? What does the State mention regarding the legal recognition of children of LGBTIQ+ families?

To approximate an answer, from documentary analysis, it is possible to understand the representations that are constituted in the political action of the State in relation to LGBTIQ+ families, in terms of presence or absence, which would allow us to discover the *discursive facts* and part of the *putting into action* of the discourses [33] around these families, while understanding how the problem that public policies are trying to solve is being represented [34]. Thus, an approach to Foucault's proposal regarding governmentality functions as the theoretical support for this research.

### 1.2. Theoretical Aspects

Following Foucault [33], the crystallization of the paradigm of the heterosexual, cisgender, monogamous, legitimate, and procreative family—and its diversification into different *devices of sexuality*—is a theme that penetrates strongly into the processes of *subjectivation of subjects*, especially in the field of sexual diversity. This idea had its starting point in the formalization of the family in the context of the eighteenth-century Enlightenment. There, the comprehension of residence and kinship, in relation to marriage [24], moved to be configured as a nuclear element of the current neoliberal economic system [35]. In this way, the family, as we understand it today, absorbed what the State cannot provide [36].

Thus, the State focuses on the family as a central axis of articulation of different rationalities that materialize in rights and duties, and in a whole public apparatus that is constituted for, and by, the family [36]. This is where certain types of relationships—not recognized within "a family"—are observed as unintelligible and therefore ungovernable and may be excluded from different political actions [3] [37].

This research considers the analytical approach to governmentality [38–40] that corresponds to a strategic field of power relationships (not only of political power), understanding them as mobile, transformable, and reversible. They involve the government of

self and others, and forces of resistance [38]. Foucault describes three senses of governmentality: (a) one as the set of institutions, procedures, analysis, reflections, calculations, and tactics that allow power to be exercised over the population (as knowledge) and as the political economy (technical instrument and security device); (b) as a force "of government" over sovereignty and discipline, the expression of which lies in specific apparatuses of government and knowledge; and (c) as a result (and process) from the justice state of the Middle Ages to the governmentalized administrative state [39]. From this notion of Foucault, it is possible to understand two perspectives [41]. One as *an object of research*, from where the analysis of government techniques from political rationalities has emerged; and another as *an analytical tool*. In this second meaning, it is possible to distinguish levels of analysis, methods, and periodization of each, incorporating the analysis of government techniques and political rationalities, together with the processes of subjectivation. Thus, as an analytical perspective, it allows the formulation of specific questions about situations that try to understand it, which can be thought from empirical research [40], for example, the objective of this study: *to understand how filiation is represented in the State and how this affects LGBTIQ+ families.*

## 2. Method

As a methodological strategy, an approach to documentary analysis is used from the proposal of post-structuralist analysis of public policies, *WPR: What's the problem represented to be?* by Carol Bacchi [42]. In this process, it is understood that the documents are a materialization of the political action of the State and that it has an impact on the processes of subjectivation of the individuals. Therefore, the documents that were selected, through *selective sampling* [43], correspond to official writings of the State, i.e., material prepared by State institutions, and they are freely available (for example, in the Library of the National Congress of Chile [BCN], on the websites of the ministries, etc.). Additionally, information obtained through Chile's law on Access to Public Information [44], known as the "transparency law", which allows information to be requested directly from State institutions, was considered.

Documents that present mentions, conceptualizations, or discussions alluding to filiation in three *local centers of experience* (following the Foucauldian nomenclature) were considered, namely: (a) Assisted Reproduction Program of the National Health Fund (FONASA); (b) Adoption office of the National Service for the Specialized Protection of Children and Adolescents (Mejor Niñez); and (c) Chilean Civil Registry. The types of materials selected were:

1. Legal documents: laws and legal advice for members of the Chilean congress.
2. Official guides of State institutions.
3. Official websites of State institutions.
4. Responses issued by government institutions through transparency law.

From the initial review of bibliographic material, 50 documents and archives were evaluated and selected. In the second stage, the material was reviewed in depth, applying the aforementioned criteria. This process resulted in the selection of 28 documents, in coherence with the documentary analysis methodology, going from a superficial reading to an in-depth reading to finally creating interpretations [45].

It is worth mentioning that by applying selective sampling in the documentary analysis, it is intended, to some extent, to represent the complete body of archives and documents that constitute the object of study [45]. However, this representativeness in no case obeys statistical criteria but is oriented to the problem in question and may imply certain selection biases, considering the available material in the current context.

The in-depth analysis of the documents was developed based on categories from the theoretical framework presented [34]. The documents analyzed are presented below (see Table 1).

**Table 1.** Documents analyzed.

| | I. Documents FONASA Assisted Reproduction Program | |
|---|---|---|
| **No** | **Document Name** | **Reference** |
| 1 | Chile Atiende: Programa de fertilización asistida de baja y alta complejidad en la red pública de salud [Chile AtIende: Assisted Fertilization Program of Low and High Complexity in the Public Health Network]. | Chile Atiende. (12 de septiembre de 2022). Programa de fertilización asistida de baja y alta complejidad en la red pública de salud. https://www.chileatiende.gob.cl/fichas/23778-programa-de-fertilizacion-asistida-de-baja-y-alta-complejidad-en-la-red-publica-o-red-preferente-mai-de-fonasa#:~:text=El%20Fondo%20Nacional%20de%20Salud,de%20instituciones%20privadas%20en%20convenio. |
| 2 | Página web FONASA. Programas Especiales: Fertilización Asistida [FONASA website. Special Programs: Assisted Fertilization]. | Fondo Nacional de Salud. (s.f). Programas Especiales: Fertilización Asistida. https://www.fonasa.cl/sites/fonasa/beneficiarios/programas-especiales. |
| 3 | Biblioteca del Congreso Nacional de Chile. Informe de Asesoría: Tratamientos de Fertilidad Asistida y FONASA [Library of the National Congress of Chile. Advisory Report: Assisted Fertility Treatments and FONASA] | Biblioteca del Congreso Nacional de Chile. (2014). Informe de Asesoría: Tratamientos de Fertilidad Asistida y FONASA. Biblioteca del Congreso Nacional de Chile. |
| 4 | Biblioteca del Congreso Nacional de Chile. Informe de Asesoría: Maternidad subrogada [Library of the National Congress of Chile. Advisory Report: Surrogacy] | Biblioteca del Congreso Nacional de Chile. (2019). Informe de Asesoría, Maternidad subrogada: Regulación en algunos países donde está permitida. Biblioteca del Congreso Nacional de Chile. |
| 5 | Subdpto. De transparencia y ley de lobby fonasa Respuesta solicitud de información No AO004T0004754 en relación al acceso de familias homoparentales a tratamientos de fertilización asistida de baja complejidad. [sub-department of transparency and lobby law fonasa Response request for information No AO004T0004754, regarding access protocols to assisted fertility treatments and it's availability for LGBTQ+ families] | Subdepartamento de Transparencia y Ley de Lobby FONASA (16 de mayo de 2022a). OFICIO ORDINARIO 1K N° 7867/2022, RESPUESTA SOLICITUD DE INFORMACIÓN No AO004T0004754. [Archivo PDF]. |
| 6 | Subdpto. De transparencia y ley de lobby fonasa Respuesta solicitud de información No AO004T0004800 en relación al acceso y requerimientos para parejas lesbomaternales, homopaternales y mujeres solteras al Programa de Reproducción Assistida de FONASA. [sub-department of transparency and lobby law fonasa Response request for information No AO004T0004800, regarding the specific requirements and access possibilities for lesbian couples, gay couples and single women to FONASA's assisted reproduction program] | Subdepartamento de Transparencia y Ley de Lobby FONASA (16 de mayo de 2022b). Oficio ordinario 1k n° 7869/2022, respuesta solicitud de información No AO004T0004800. [Archivo PDF]. |
| 7 | FONASA Código de Ética [FONASA's (National Health Fund) Ethic Code] | Fondo Nacional de Salud (2017). Código de Ética. https://www.fonasa.cl/sites/fonasa/documentos. |
| 8 | MINSAL Guía para el Estudio y Tratamiento de la Infertilidad. [Ministry of Health's Guide for the study and treatment of infertility] | Programa Nacional Salud de la Mujer. (2015). Guía para el Estudio y Tratamiento de la Infertilidad. http://www.repositoriodigital.minsal.cl. |
| 9 | Diario Oficial modifica resolución exenta No 277/2011. Donde el Ministerio de Salud establece criterios de calidad y cálculo que se utilizarán para evaluar instituciones prestadoras de técnicas de reproducción asistida. ["Diario Oficial amends exempt resolution No. 277/2011. Where the Ministry of Health establishes quality and calculation criteria that will be used to evaluate institutions that provide assisted reproduction techniques.] | Diario Oficial. (2019). Modifica resolución exenta no 277/2011 del ministerio de salud, que aprobó las normas técnico administrativas para la aplicación del arancel del régimen de prestaciones de salud del libro ii dfl no 1, de 2005, del ministerio de salud, en la modalidad de libre. https://www.diariooficial.interior.gob.cl/publicaciones/2019/05/22/42359/01/1592605.pdf. |

**Table 1.** *Cont.*

| No | Document Name | Reference |
|---|---|---|
| **I. Documents FONASA Assisted Reproduction Program** | | |
| 10 | BCN<br>Ley nº 21.400<br>Modifica diversos cuerpos legales para regular, en igualdad de condiciones, el matrimonio entre personas del mismo sexo.<br>[It modifies various legal bodies to regulate, under equal conditions, marriage between people of the same sex.] | Ley nº 21.400. (2021). Ministerio Secretaria General de Gobierno. http://bcn.cl/2ucii |
| **II. Documents Civil Registry and Identification Service** | | |
| | **Document Name** | **Reference** |
| 11 | Servicio de Registro Civil e Identificación<br>Formulario de Acuerdo para establecer el orden de los apellidos, firmado por ambos progenitores (C-9)<br>[Service of civil registration and ID<br>Agreement Form to establish the order of the surnames, signed by both parent (C-9)] | Servicio de Registro Civil e Identificación. (2022). Formulario C-9: Acuerdo de los padres/progenitores para la determinación en el orden de transmisión de los primeros apellidos a sus hijos comunes menores de edad. [Archivo PDF]. |
| 12 | Página Web Servicio de Registro Civil e Identificación<br>Documento informativo sobre cómo realizar Solicitud para Hora de Matrimonio<br>[Civil Registry and Identification Service's web page<br>Informative document on how to make a Request for Marriage appointment] | Servicio de Registro Civil e Identificación. (2021). Solicitud hora matrimonio. https://www.registrocivil.cl/principal/canal-tramites/solicitud-matrimonio-igualitario-2. |
| 13 | Respuesta a solicitud de información AK002T0020968. Se consulta sobre los procesos y particularidades de la inscripción de hijos por parte de familias LGBTQ+.<br>[Response to request for information AK002T0020968. The processes and particularities of the registration of children by LGBTQ+ families are consulted.] | Unidad de Transparencia y Sistema de Integridad del Servicio de Registro Civil e Identificación (03 de mayo de 2022). Requerimiento de información número AK002T0020968. [Archivo PDF]. |
| 14 | Respuesta a solicitud de información AK002T0021096. Se consulta sobre los procesos y particularidades tanto del proceso de matrimonio como de inscripción de hijos por parte de familiar LGBTQ+.<br>[Response to request for information AK002T0021096. The processes and particularities of both the marriage process and the registration of children by an LGBTQ+ family member are consulted.] | Unidad de Transparencia y Sistema de Integridad del Servicio de Registro Civil e Identificación. (01 de junio de 2022) Requerimiento de información número AK002T0021096. [Archivo PDF]. |
| 15 | Noticia La Tercera<br>Sobre Ley de Filiación y los 20 años desde la promulgación de la ley que terminó con la consideración de "hijos ilegítimos" en Chile.<br><br>[La Tercera's News about the 20-year anniversary since the law that ended the existence of "illegitimate children" in Chile.] | Sepúlveda, P. (1 de Julio de 2018). A 20 años de la ley que terminó con los hijos ilegítimos en Chile. Obtenido de La Tercera: https://www.latercera.com/tendencias/noticia/20-anos-la-ley-termino-los-ninos-ilegitimos-chile/227203/ |
| 16 | Biblioteca del Congreso Nacional (BCN); Sobre Ley de Filiación<br>[Congress National Library (BCN); Legal guide about Filiation Law] | Biblioteca del Congreso Nacional de Chile. (8 de Septiembre de 2022). Guía legal sobre: Filiación. Obtenido de BCN: https://www.bcn.cl/leyfacil/recurso/filiacion |
| 17 | Página web Senado<br>Sobre filiación en parejas del mismo sexo<br>[Senate's web page; Informative about filiation in same sex couples] | Senado Chile. (19 de Julio de 2021). Filiación de hijos e hijas de parejas del mismo sexo: avanza la votación de indicaciones. Obtenido de Senado Chile: https://www.senado.cl/noticias/filiacion/filiacion-de-hijos-e-hijas-de-parejas-del-mismo-sexo-avanza-la-votacion |

**Table 1.** *Cont.*

| I. Documents FONASA Assisted Reproduction Program | | |
|---|---|---|
| **No** | **Document Name** | **Reference** |
| 18 | Artículo de revista<br>Sobre inicios del Registro Civil en Chile y la ruptura con la Iglesia<br>[Journal Article about the beginnings of Chile's National Registry and it's tensions with the Church] | Irarrázaval, A. (2014). Los inicios del registro civil de Chile: ¿Ruptura o continuidad con las antiguas partidas eclesiásticas? Revista de estudios histórico- jurídicos(36), 315-341. Disponible en: https://www.scielo.cl/scielo.php?script=sci_arttext&pid=S0716-54552014000100011 |
| 19 | Tesis de Pre grado/Memoria<br>Ciencias Jurídicas sobre la evolución de la filiación.<br>[Undergraduate Thesis of Legal Sciences on the evolution of filiation.] | Gajardo, A. (2009). La filiación: un análisis de su evolución. Santiago: Universidad de Chile. Disponible en: https://repositorio.uchile.cl/handle/2250/106977 |
| III. Mejor Niñez [Better Childhood] Documents on Adoption | | |
| | **Document Name** | **Reference** |
| 20 | BCN<br>Ley nº 19.620, la cual dicta las normas sobre la adopción de menores de edad en Chile.<br>[Congress National Library, Law No. 19,620, which dictates the rules on the adoption of minors in Chile.] | Ley nº 19.620. (2021). Dicta normas sobre la adopción de menores. Ministerio de Justicia. |
| 21 | BCN<br>Ley nº 21.400, la cual modifica diversos cuerpos legales para regular, en igualdad de condiciones, el matrimonio entre personas del mismo sexo.<br>[Congress National Library, Law No. 21,400, which modifies various legal bodies to regulate, under equal conditions, marriage between people of the same sex.] | Ley nº 21.400. (2021). Modifica diversos cuerpos legales para regular, en igualdad de condiciones, el matrimonio entre personas del mismo sexo. Ministerio Secretaria General de Gobierno. http://bcn.cl/2ucii |
| 22 | BCN<br>Ley 2.675 sobre la protección de la infancia desvalida<br>[Congress National Library, Law 2,675 on the protection of "helpless children"] | Ley n° 2.675. (1912). Ley sobre protección a la infancia desvalida. Ministerio de Justicia. |
| 23 | Página Web Servicio Nacional de Protección Especializada a la Niñez y Adolescencia<br>[Web Page: National Service for the Specialized Protection of Children and Adolescents] | Servicio Nacional de Protección Especializada a la Niñez y Adolescencia (s.f). https://www.mejorninez.cl/. |
| 24 | Gobierno de Chile<br>Manual del Usuario Postulante, relacionado a los procesos de adopción de un niño/a delcarado/a susceptible de ser adoptado/a.<br>[Government of Chile Applicant User Manual, related to the adoption processes of children declared to be susceptible to adoption.] | Gobierno de Chile (octubre 2021). Manual del Usuario Postulante (PRODUCCIÓN). Módulo "Solicitante Nacional" - Sistema Informático Integrado de Adopción (SIIA). [Archivo PDF] |
| 25 | Servicio Mejor Niñez(a)<br>Respuesta a solicitud de información pública N° AI010T0000407. Se consulta por la regulación del proceso de adopción por parte de parejas LGBTQ+, disponibilidad de la información al respecto y diferencias respecto a parejas heterosexuales.<br><br>[Servicio Mejor Niñez(a) Response to request for public information No. AI010T0000407. It consults about the regulation of the adoption process by LGBTQ+ couples, availability of information in this regard and differences with respect to heterosexual couples.] | Mejor Niñez. (11 de mayo 2022). Solicitud de Información Pública N° AI010T0000407, ingresado en Portal de Trasparencia del Servicio Mejor Niñez. [Archivo PDF]. |

**Table 1.** *Cont.*

| I. Documents FONASA Assisted Reproduction Program | | |
|---|---|---|
| **No** | **Document Name** | **Reference** |
| 26 | Servicio Mejor Niñez(b)<br>Respuesta a solicitud de información pública<br>N° AI010T0000416. Se consulta sobre los requisitos del proceso de adopción en general y la existencia de parejas del mismo sexo que hayan logrado concretar una adopción desde la promulgación de la ley de matrimonio igualitario (N 21.400).<br><br>[Servicio Mejor Niñez (b)<br>Response to request for public informationNo. AI010T0000416. The requirements of the adoption process in general and the existence of same-sex couples who have managed to finalize an adoption since the promulgation of the equal marriage law (N 21.400) are consulted.] | Mejor Niñez. (24 de mayo 2022). Solicitud de Información Pública N° AI010T0000416, ingresado en Portal de Trasparencia del Servicio Mejor Niñez. [Archivo PDF]. |
| 27 | Página Web Programa Chile Crece Contigo<br>Homoparentalidad y crianza: Madre y padre como una función.<br><br>[Chile Crece Contigo's (Public Institution that promotes healthy childhood) take on "Same sex couples and upbringing: Mother and father as a function."] | Subsecretaría de la Niñez (s.f). Homoparentalidad y crianza: Madre y padre como una función. Chile Crece Contigo. Protección Integral a la Infancia. https://www.crececontigo.gob.cl/columna/homoparentalidad-y-crianza-madre-y-padre-como-una-funcion/. |
| 28 | Página Web Programa Chile Crece Contigo<br>Mirando la homoparentalidad y la lesbomaternidad desde la crianza<br><br>[Chile Crece Contigo's (Public Institution that promotes healthy childhood) take on the upbringing or raising children process by same sex couples] | Subsecretaría de la Niñez (s.f). Mirando la homoparentalidad y la lesbomaternidad desde la crianza. Chile Crece Contigo. Protección Integral a la Infancia. https://www.crececontigo.gob.cl/columna/mirando-la-homoparentalidad-y-la-lesbomaternidad-desde-la-crianza/. |

This proposal allows an interpretation of the conceptual forms through which a particular problem is thought and how they are reflected in public documents of the State. These conceptualizations allow the interpretation of different ways of understanding the problem. In short, the research team proposed an analytical judgment that allowed the demonstration of the naturalization of the problem enunciated in the public documents of the Chilean State [46], showing how this situation affects LGBTIQ+ families.

## 3. Analysis

### 3.1. "Mejor Niñez" and the Adoption Process

In Chile, from a historical point of view, the problem underlying policies related to the adoption of children is initially represented under the Law for the Protection of Helpless Children of 1912, being the first regulation that was enacted to solve the situation of parental abandonment, child abuse and some forms of child exploitation. Although its application was very discreet, it marked the beginning of a State policy oriented towards the protection of the rights of children and young people, and adoption was later institutionalized by Law No 5.343 in 1934 [47].

Currently, the National Service for the Specialized Protection of Children and Adolescents, also known as Mejor Niñez [Better Childhood], is responsible for "Protecting and restoring the human rights of children, adolescents, and young people seriously threatened or violated in their rights" [48]. Mejor Niñez is the successor of the National Service for Minors (SENAME), which received numerous criticisms for the systematic rights violation of children and young people in its care [4].

Mejor Niñez begins operating in October 2021, with adoption being one of the main axes of work, on which it is declared:

> This line of action will seek a family for the child or adolescent, *whatever their composition* [the highlight is their own]. The idea is that this family gives him affection and seeks care that satisfies his bonding and material needs, when this cannot be provided by his family of origin. [49] par. 1

The regulations governing the work of Mejor Niñez in relation to adoption come from Law No. 19.620 (Child Adoption law) which was last amended in 2021, in which the objective of adoption is:

> ( . . . ) to ensure the best interests of the adoptee, and to protect his right to live and develop within a family that provides him with affection and care to satisfy his spiritual and material needs, when this cannot be provided by his family of origin. [50] Art. 1

In this way, the law defines as susceptible to adoption those who are in the care of people who do not want or cannot take charge of their care, whether physically, morally, or economically. On the other hand, it is defined as *capable of adopting* persons over 25 years of age and under 60 who have a minimum difference of 20 years with the child or adolescent they will adopt (except for direct relatives) where "Adoption may be granted to Chilean or foreign spouses, with permanent residence in the country, who have two or more years of marriage" [50] Art. 20. It emphasizes that this last requirement will not be required if one or both spouses are infertile. Finally, only if a child or adolescent, "susceptible to being adopted", does not have "interested spouses", a *single, divorced, or widowed* person may choose, thus demonstrating a hierarchical model of Victorian family suitability, an inheritance that resists perishing in the face of new family models. In this regard, those who wish to adopt must certify their suitability by being evaluated "physically, mentally, psychologically and morally by SENAME or an organization authorized to carry out adoption programs" [49].

Thus, in the problem represented by the current legislation, and framed in the socio-historical context briefly mentioned, children and adolescents who choose to be adopted are understood as vulnerable subjects, whose stay under the guardianship of the State [5] is transitory and is framed in the following cases: first, that the State will be a better rights guarantor than their original families and that the State institution or authorized by the State will have the power to define morally the suitability of a family, from expert knowledge, to maintain the guardianship of their child or a family they wish to adopt. These power-knowing dynamics show rationalities based on traditional family models, showing a position that makes LGBTIQ+ people who wish to become parents invisible [5,14].

These assumptions come into tension since a neutral perspective is not possible in relation to a moral judgment, which will inevitably be influenced by the sociocultural, political, and religious context in which discrimination against LGBTIQ+ people has been built, i.e.: heteronormativity [1,14,51,52].

In this line, the moral judgment in Chile that defines the suitability of a family is linked, in one way or another, with heteronormative discourses, reflected in studies on public discourses that are socially expanded and that consider heterosexual couples more suitable to provide care to children and adolescents [22,53,54]. In this way, the heteronormativity permeates the mentalities and representations of the State, building a whole political apparatus, in this case in terms of filiation, which reproduces stigmas and a system of permanent exclusion for LGBTIQ+ families.

In this regard, the discussion in the Chilean congress, during the first constitutional procedure of Law 21.400, is an excellent reflection of hate speech that influences the processes of the construction of public policies. For example, Senator Luz Ebensperger emphasizes that "( . . . ) We need a father and a mother!; we need society and humanity to continue to advance in marriage between a man and a woman; We need the priority goal of the former to remain procreation" [55] (p. 87).

Despite this, with the entry into force of the Marriage Equality law [18], it is established that marriages formed by persons of the same sex could "adopt on equal terms with

heterosexual marriages" [56]. The discussion on this section of the law was the focus of heteronormative discourses, as mentioned by the same senator "The right of two people to marry cannot be prioritized over the right of children to know their biological father or mother" [55] (p. 257).

In relation to same-sex couples who are living together, in the two responses delivered in May of 2022 through the transparency portal [56,57], Mejor Niñez states that current legislation does not allow joint adoption, being able to adopt only one of them, although both are evaluated, i.e., due to the current legislation, in the absence of modifications in the adoption law, same-sex couples must dissolve their civil union agreement (if they have one) to begin a process—as a single person—since the law requires a minimum of two years as married couples or, the possibility of adopting as single people framed in the aforementioned hierarchy dynamic: Unclear but undeniably present.

And in the latter case, even if one of the members of the couple adopts, the two will be evaluated "Because, by the fact of living together, both will constitute significant figures for the child" [57]. However, the legal filiation bond of the adopted child or adolescent will be established only with one of the cohabitants [56].

In this situation, governmentality is expressed from a heteronormative position that affects people's forms of organization. In addition, that particularly affects the understanding of LGBTIQ+ subjects as subjects of another level of rights and, therefore, of people. The representations that can be read from the absences in public policies, thought from heteronormative logics, constitute subjects that seem to be unintelligible, even when the laws nominate them explicitly. Thus, following Bacchi [42], we can see how, although the issue of filiation in LGBTIQ+ families can be thought of from the State, the practical operation of the norm is restricted by another group of norms that constrain them, accounting for the *tactical polyvalence of the discourses*, in Foucauldian terms [33]. That is: it is evident how the forms of power diversify in trying to maintain a given order from different perspectives.

In addition to the gap to establish recognized filiation bonds for unmarried LGBTIQ+ couples, or united through the civil union agreement, it is clarified that due to the entry into force of the Equal Marriage law less than two years ago (from 10 March 2022), marriages formed by same-sex couples would not meet the criteria for joint adoption until March of 2024 [57]. Finally, and paradoxically, it is emphasized that "the sexual orientation of applicants for adoption is not recorded, since the current regulations do not make any distinction in their regard" [56].

Thus, understanding that the Equal Marriage law entered into force in March 2022 and opened the possibility of same-sex couple's adoption, it would be expected that there would be public guidelines that will accompany the process as well as transitory measures that would allow adoption before the passage of two years by couples who, in line with the argument of the requirement, demonstrate cohabitation or civil union agreement during the required period. In this regard, the documentary analysis shows the scarcity of information related to adoption by same-sex couples, with a significant gap in access to information and hindering the possibility of legal filiation by LGBTIQ+ families. In addition, there is no mention in any document of elements that guide the adoption processes of families formed by trans, intersex, or other communities that do not correspond to cisgender gay or lesbian couples.

In short, the State practices visualized in the documents are characterized by a silencing of the formation of LGBTIQ+ families through adoption processes, there are legal gaps (such as the requirement of 2 years of marriage), as well as sociocultural and religious, such as the relationship of the Catholic Church with the accredited institutions for the delivery of the certificate of "suitability", which, although not made explicit, have historically positioned cisgender heterosexual families at a higher hierarchical level to the detriment of the right to form families of LGBTIQ+ people and of children and adolescents who could opt for filiation ties with such families [16].

Although the Equal Marriage law lays the foundations for changes in the adoption processes by LGBTIQ+ families, the ideal of the family valued hierarchically by economic,

social, cultural, religious, and sex-generic orientation elements is perpetuated under a heteronormative perspective that translates into the materialization of regulations under power-knowing logics.

*3.2. FONASA Assisted Fertilization Program*

The FONASA Assisted Reproduction Program provides coverage for assisted reproduction procedures of high and low complexity, responding to what is described as a global problem: infertility [58]. Assisted reproduction is defined as a "( . . . ) medical treatment based on a set of procedures that seeks through different assisted reproduction techniques, to facilitate pregnancy in those people who for various reasons cannot achieve it" [59]. The program is divided into two categories: low and high complexity, and must exhaust the possibilities of the first group, before access to high complexity methods. The benefits included in these levels have 100% coverage in the public health network for people affiliated with FONASA. In institutions of the private health network that have an agreement with FONASA, payment is made through the purchase of a voucher (Free Choice Modality), with variable values depending on the sex of the person receiving the benefit and the level of complexity.

Thus, the inclusion criteria to qualify for the low-complexity program are "12 months of unprotected sexual relations" in the public network and medical indication of infertility "of both men and women" [60] in the private network. The above suggests the problem represented: *medically induced infertility of cisgender heterosexual couples.*

Although within the description of the program its exclusivity towards heterosexual couples is not explicit at any time, the inclusion requirements described allow us to infer that politics and the problem represent a rationality expressed towards the construction of heterosexual and cisgender subjects. The justification for the creation of such a program lies in the socio-demographic analysis, where it is argued that:

> "The fertility rate at the country level reflects a dramatic decline in the number of children per woman in recent decades. In addition, it is evident that pregnancies are occurring at older ages of women, where the risk of infertility increases". [60] par. 1

Thus, infertility is understood from that representation: based on the decrease of children per woman and the age at which they gestate (it is interesting to note the constant omission of the role of heterosexual men in the problem). However, the explicit justification of public policy is not restricted to the field of health, making specific mentions of other aspects of the problem such as mental health and the social exclusion of heterosexual couples who cannot have children. This is clearly expressed in the following quote: "For many couples, the impossibility of having children is seen as a loss, a failure and the feeling of social exclusion, pressure that increases the risk of separation and divorce" [58] (p. 4). Here, once again the heteronormative idea of the Victorian family that remains married until the end of their days and maintains an avid offspring materializes [33]. In the same vein, FONASA states that "the Government is promoting measures that help couples who wish to do so, to fulfill their desire to have a child" [60]. The representation of the problem in this program could understand infertility as a psychosocial phenomenon that is addressed through a State that "promotes and supports desired motherhood" [60]. In coherence with the above, a comprehensive response to the problem (with social, political, psychological, and medical edges) and inclusive would be expected; however, the benefits included in the Assisted Reproduction Program are limited to the heteronormative biomedical approach: the diagnosis of infertility and assisted reproduction procedures such as assisted fertilization for heterosexual couples.

The linear association made by the program between a decrease in births and infertility is striking, since, as Zegers et al. mention, "the decline in the birth rate in Chile is not the direct result of an increase in infertility, but there are sociocultural changes of great significance" [61] (p. 26).

In this regard, a widely discussed subject corresponds to the definition of the concept of infertility, which in 2009 is defined by the WHO as a disease of the reproductive system characterized by the inability to achieve a clinical pregnancy after 12 months of unprotected sexual relations. Subsequently, following discussions by international organizations, the definition is updated by adding to unprotected relationships the impediment in a person's ability to reproduce, either as an individual or with his partner [62].

Although the phenomenon of infertility has been present throughout history, the demarcation of this problem in the biomedical field responds to the development of genetic techniques and the important scientific progress of the last century. Thus, the biological and pathological component takes a more important role in the justification of the problem. This represents the Foucauldian idea of a device for the diversification of knowledge about sexuality and reproduction as a form of power over bodies [33].

This arbitrariness tries to be justified from the supposed impossibility for LGBTIQ+ people to procreate in a "natural" way. Legally, the same criterion does not apply when couples made up of heterosexual people perform adoption procedures and/or assisted reproduction techniques. This gap suggests that discrimination is not based on the (non)existence of a blood relationship between children and parents, but rather, on the sexual and affective orientation of the couple, which is opposed to the recognition of their human rights. In this sense, vulnerabilities that transversalize the diverse identities of LGBTIQ+ people are reinforced and perpetuated through government policies, laws, and programs [63].

On the other hand, one of the representations of the problem is the breakdown of the family institution, expressing in the Assisted Reproduction Program the risk of separation or divorce. If we consider that the material of the program analyzed is before the enactment of the Equal Marriage law, it becomes evident that the risk of separation and divorce refers to heterosexual couples. Within the above, the requirements to qualify for the program refer to the idea that "You and your partner do not need to be legally married, but you must demonstrate that you live in a relationship of social, affective, and stable family coexistence, for at least 2 years" [60]; however, the criteria of medical infertility continue to leave the LGBTIQ+ community out of the target population.

The arguments that support the policy explicitly are based on the biomedical approach and a post-positivist paradigm, with specific mentions of the social implications of the problem, but without incorporating these into the model of response to this representation. This is considered a reductionist approach to complex phenomena such as human reproduction. In this line, infertility in heterosexual couples also frames the impossibility of having children as a personal responsibility, attributable only to some people and "solvable" only when entering into certain characteristics, where in addition to the sex-affective orientation an important socio-economic gap stands out: despite the existence of a program with State funds, the FONASA Assisted Reproduction Program does not represent more than 15% of reproduction cycles assisted at nationwide [64].

Thus, the declaration of infertility as a social problem and a concern for the mental health of couples is futile and restricted to the hegemonic group that is included in the heteronormative family construct. On the other hand, the Assisted Reproduction Program refers to the objective of supporting the desired motherhood, a principle that apparently would refer exclusively to the heterosexual family as an institutionality "organizing society" [65].

In this line, an underlying assumption we find is the idea of the desire to have children as a right or unique attribute of heterosexual couples, where in relation to filiation, the conjugal family confiscates it. In addition, it absorbs entirely the seriousness of the reproductive function [33]. The program omits the existence of other types of families along the LGBTIQ+ spectrum.

This produces a false scenario in which they intend to cover the needs associated with family planning without taking real and inclusive actions (not even for heterosexual couples, since coverage is not complete and does not include associated complications). A clear interruption to this construction of the problem must arise from the recent recog-

nition of the rights of LGBTIQ+ families, since as Mondaca et al. refer, "The obstacles related to the exercise of maternity/paternity from sex-affective diversity generate affective self-limitations associated with the (im)possibility of thinking of oneself as a potential mother/father based on the hostile social context they face" [1] (p.11).

*3.3. Registration of Children in the Civil Registry*

This third case analyzed corresponds to the filiation process in charge of the Civil Registry of Chile. Filiation is the relationship of descent that exists between two people, one of whom is the father or mother of the other, and can be granted by nature, by assisted human reproduction techniques, and/or by adoption [66].

The principles of filiation are equality of all children, non-discrimination on grounds of birth, and the supremacy of the best interests of the child, which emphasizes their consideration as subjects of law. It also establishes that everyone has the right to know his biological origin and to belong to a family. It also determines that the law must recognize equal rights to both children born out of wedlock and those born within it [66].

Following the enactment of the Marriage Equality Act, the Special Commission responsible for processing bills relating to children and adolescents initiated a motion regulating the right of filiation of children of same-sex couples. This is when the concept of parents is defined, being established that co-parenthood will be called the filiation determined with respect to two fathers or two mothers [55].

This instance also recognizes the reproductive autonomy of the person, which includes the right to find a family and equal access to the technology necessary for this purpose, prohibiting the conditioning of access to it for any reason, including the lack of stability in the couple, a certain sexual orientation or diagnosis of infertility. In addition, it makes the rules on adoption applicable to co-paternal and co-maternal couples [55].

Through the review of documents on filiation in Chile, an analysis is made of the declared process of the registration of children by same-sex couples and how the Civil Registry responds to this. Therefore, we observe that in the language used, there is still a restrictive definition of the relationship of filiation, where the biological link with a mother and/or a father prevails: "If it is one of the parents who recognizes, he will not be obliged to express the person in whom or of whom he had the child" [67] (p. 3).

In the response issued by the institution, to the request for information, it is recognized that it is pending to modify the vocabulary in favor of one that is in harmony with the Marriage Law [67]. In these, the concept of the parent is not reduced to a solely biological link between mother, father, and the person with respect to whom they have a relationship of filiation, since "The laws or other provisions that refer to the expressions father and mother, or father or mother, or other similar, shall be understood to apply to all parents, without distinction of sex, gender identity, or sexual orientation, unless the context or express provision must be understood otherwise" [67] (p. 3).

It is pertinent to question how and why changes to the legal processes of filiation occur since the Equal Marriage law was enacted, i.e., what is implied by what is modified in the current registration process of children to comply with the said law harmoniously and coherently. These changes show that before Law 21.400 the figure of filiation represented a restricted subject in its definition, i.e., following heteronormative aspects, which did not give space for other families to exist, both in the concrete and legal reality and in the subjectivity of LGBTIQ+ families.

In the first place, the changes to the norms to establish filiation in Chile are generated because there is a conception established before Law 21.400, where parenthood is understood from a logic of the biological bond of a heterosexual couple, still associated with the marriage bond, as a model from the State, and even the Church. Therefore, this is what is governed and what is allowed in the institutionality until then.

Second, considering the lack of information available on the filiation regulations for LGBTIQ+ families and the ongoing process of changing the language in the procedures

demonstrated in the documents [67], there is a striking lack of urgency from the State to accommodate the processes of registration of children by LGBTIQ+ families.

This change in the Civil Registry norm is beginning to generate processes of change in relation to the subjectivation of LGBTIQ+ people to visualize themselves forming families within the institutionality, but the lack of clarity and availability of information in the protocols to be followed, as well as the aforementioned contradictions reduce this right already assumed since the Equal Marriage law. Among the aspects that remain unaddressed in this regulation, we can mention the processes necessary for the particularity that each family requires to form the filiation bond, such as unilateral maternity and paternity or co-parenting or co-maternity.

Diversity in the composition of families and parentage relationships is beginning to be accommodated. In this process, there is a change in the law that follows most clearly in terms of clarity of processes, availability to the public, and coherence in the language used. At the moment there are obstacles in the symbolic, institutional, and factual spheres, which are related to the exercise of parenthood from sex-affective diversity, which can generate affective self-limitations associated with the (im)possibility of thinking of oneself as a potential mother/father [5,51].

## 4. Discussion

This research is in line with similar ones that have exposed the heteronormative representations of subjects in public policies [13,14,16,68]. In this regard, other studies on the state of public policies on sexual diversity in Chile have reported heteronormative structures to think about sexuality and bodies, which limits the possibilities of agency of the subjects [11]; or how sexual diversity itself is homogenized, generating that the difference is amalgamated and therefore not represented in its entirety [13].

These works show how the constructions of the problems that the State intends to solve are built based on the normalization and standardization of heterosexuality as a regime of social organization [69–73], i.e., the rationality of the State would be based on the understanding of heterosexuality as a normative and regulatory aspect of society and, particularly in this case, of the family.

This is related to a traditional conception of family that seems to prevail more strongly in countries where a conservative State has prevailed, as in the case of Chile. This is favored by the agency capacity of religious organizations such as private adoption foundations that manage to install their vision and criteria on what it means to be a family, and how it should be composed [74].

Along these lines, Warner proposed how heteronormativity can be related to:

"The family, with the notions of individual freedom, the State, public expression, consumption and desire, nature and culture, aging, reproductive politics, racial and national fantasy, class identity, truth and trust, censorship, intimate life and social display, terror and violence, health care, and deep cultural norms about body bearing". [70] (p. 6)

Thus, it is understood that heteronormativity is at the basis of every understanding of the actions of the State and, therefore, this constitutes a crucial support for the configuration of governmentality. In this way, around the approval of specific laws for LGBTIQ+ families, the Chilean State has sought an intermediate point in the logic of the least possible disruption. Thus, public policies that emanate from the pressures of civil society in matters of sexual diversity are approved under the notion of the "middle ground", when in terms of human rights, it is not possible to think under this logic. Another iconic example is the civil union agreement and how, by excluding the filiation of same-sex couples, it negatively affected couples who wished to be parents [5]. Depriving them of the right to form a family.

Particularly in the field of governmentality, several investigations that have focused on the Chilean State have centered their analysis on institutions that work with families, pointing out the imaginaries around their construction from neoliberal perspectives, reporting programs aimed at families where the ideas of individualism, merit, competence,

entrepreneurship, integration, and consumption prevail [75] and falling into the risk of naturalizing their structures, where gender, social class, nationality, among other aspects, are represented in public policy as equals [76], i.e., it is thought of as equality of conditions (free market) to access benefits or rights that the State could provide. However, from the documentary analysis, we see that these investigations are not crossed by an axis that calls into question the heteronormativity with which these programs are built, where a biological logic still prevails, which determines in a patriarchal axis the descent and how the construction of the family is thought, where some families can access certain benefits that others cannot. In other words, a heteronormative construction "of the problem" impacts public policy by excluding LGBTIQ+ families from their reproductive rights in the formation of families.

Other studies have focused on government tactics, expressions of governmentality, and the process of working with families. From this logic, disciplinary norms are maintained, as an exercise of power-knowledge, which materialize in direct work with families, subordinating them to the State's rationalities [77]. In this sense, although this research did not address the direct work between the State and LGBTIQ+ Families, we wondered how these rationalities could affect the subjectivities of these families: thinking of themselves as mother(s) or father(s) and their link with the rationalities of Chilean and Latin American public policies of filiation.

## 5. Conclusions

In the different local centers of experience analyzed, it is possible to identify its correlation with the Latin American situation. Particularly in relation to its/their limited access to assisted reproduction [64,78]; moral judgments regarding the suitability of LGBTIQ+ families to adopt and/or establish filiation bonds [79–81], among others.

In this way, the description of the framed situation in Chile dialogues with other studies of the territory, building a Latin American knowledge of filiation in LGBTIQ+ families, with a view aiming at contributing to an action of the State consistent with the sexual and reproductive rights of their population.

Finally, regarding the construction of public policies, silence stands out as a form of exclusion present in the rationalities of the three local centers of experience studied: the Mejor Niñez adoption program, the FONASA Assisted Reproduction Program, and the Civil Registry, it being necessary for future studies to investigate the role they play in the construction of subjectivities in LGBTIQ+ families.

It is concluded that a broader and more diverse understanding of such represented problems would contribute to a greater representation of diversity in State policies. It is considered that the present study contributes to making visible how certain discourses that come from the State and policies are installed or lowered in society, hindering the possibility of generating transformations. It contributes to the dialogue, providing arguments that show that there is discrimination against same-sex couples regarding the right to establish a family.

**Author Contributions:** Conceptualization, R.M.; Formal analysis, R.M., G.M.Y., J.H.Q., F.G.B. and P.O.-A.; Funding acquisition, R.M.; Investigation, R.M., G.M.Y., J.H.Q. and F.G.B.; Methodology, R.M.; Project administration, R.M. and G.M.Y.; Supervision, R.M.; Writing—original draft, R.M. and G.M.Y., J.H.Q. and F.G.B.; Writing—review & editing, R.M. and P.O.-A. All authors have read and agreed to the published version of the manuscript.

**Funding:** This research was funded by Agencia Nacional de Investigación y Desarrollo, (ANID) Chile, grant number: FONDECYT de Iniciación no 11220183. And The APC was funded by FONDECYT de Iniciación: 11220183: "Familias LGBTIQ+ y acción política del estado chileno: el parentesco y la filiación entre relaciones de poder y resistencia"[LGBTIQ+ Families and political action of the Chilean state: relations of power and resistance with kinship and filiation].

**Institutional Review Board Statement:** The study was conducted in accordance with the Declaration of Helsinki, and approved by the Human Subjects Research Ethics Committee [Comité de ética de Investigación en Seres Humanos] of Medicine Faculty of University of Chile [de la Facultad de Medicina de la Universidad de Chile] (Proy. No 014-2022, acta No 004; approved on 19 April 2022).

**Informed Consent Statement:** Not applicable.

**Data Availability Statement:** The material analyzed is freely available. See Table 1.

**Acknowledgments:** We would like to thank Murilo Meloto who kindly revised the first version of this writing in English. His contributions were relevant to the development of this work.

**Conflicts of Interest:** The authors declare no conflict of interest.

## Notes

1. The acronym stands for lesbian, gay, bisexual, transgender/transsexual/transvestite, intersex, queer, and others.
2. Other achievements in terms of rights for LGBTIQ+ people are the approval or modification of different laws and regulations, such as the Gender Identity Law No. 21.120 (in 2018) that *recognizes and protects the right to gender identity*; the civil union agreement Law No. 20.830 (in 2015); Anti-Discrimination Law No. 20.609. (in 2012), the modification of the article on sodomy in the penal code (in 2019), among others.
3. For example, given the indissoluble connection between marriage and family, in different representations of the Chilean State, same-sex couples would occupy this type of relationship, since, although the civil union agreement recognizes their bond, it leaves out filiation and recognition of kinship. This is a legal category that implies the recognition of the exercise of paternity/maternity in same-sex couples, as well as the recognition that their children are part of these families.
4. Thus, in a report issued by the United Nations, it was reported that 8 out of 10 children and young people belonging to SENAME reported with received physical or psychological punishment from staff.
5. In Chile, there are currently three non-State institutions authorized as "collaborating agencies" to implement adoption programs: *Fundación Chilena de la Adopción* [Chilean Adoption Foundation], *Fundación San José* [San José Foundation] and *Fundación Mi Casa* [My House Foundation]. Of these, the last two have a direct historical link with the Catholic Church [50].

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
