# Peer review of "“We Need a Father and a Mother!” Rationalities around Filiation in the State: The Invisibility of LGBTIQ+ Families"

_societies, doi:10.3390/soc13050109_

Round 1
Reviewer 1 Report
This is a very interesting paper, creative in its investigation, and addresses an important topic. The theoretical framework is clearly stated. The method however needs a better description. See my comments below.
1. Point of clarification. How are the families who are thinking about 74 these programs? Is how the appropriate word here? Is it not who?
2. How does the reader know that the documents selected are representative of all the documents in this sample? Is there a chance of skewness? This needs to be clarified so that the issue of sample bias can be adequately assessed by the reader.
3. more detail in how these documents were analysed and the data extracted and coded is required to help the reader follow the presentation of the document analysis.
4. Is it possible to insert more data into the analysis section to support the claims?
Author Response
First of all, we would like to thank you for your comments which helped us to think about various aspects of our work. Thanks for you time and dedication. Next, we will respond to each of the suggestions.
Reviewer 1:
- Point of clarification. How are the families who are thinking about 74 these programs? Is how the appropriate word here? Is it not who?
We greatly appreciate the comment. We have accepted the suggestion and changed "how" to "who" to more clearly refer to who these families are.
- How does the reader know that the documents selected are representative of all the documents in this sample? Is there a chance of skewness? This needs to be clarified so that the issue of sample bias can be adequately assessed by the reader.
Thank you very much for this comment. We have modified and added information to the methodology. By using sample selection, we specify how many files/documents were originally identified and how many were ultimately selected for analysis. We refer that they are those mentioned in the analysis, collecting the concepts of valuation, selection and disposition of the author.
Regarding the possibility of bias, it is an unavoidable threat. However, it should be considered that the documentary analysis proposed here is developed based on criteria from a qualitative research tradition, so the selection of the documents obeys the object of study, the questions that guide the research and the interest of the researcher, rather than a search for statistical representativeness. However, to carry out the selection of the documents, a rigorous analysis process was carried out based on the aforementioned criteria that allow determining which documents are selected and which are discarded and their reasons.
- More detail in how these documents were analysed and the data extracted and coded is required to help the reader follow the presentation of the document analysis.
Regarding the way in which the documents were analysed, it is worth mentioning that a general initial reading was carried out to verify whether or not they met the established criteria. Once the texts were selected, they were analysed from theoretical perspectives presented earlier in the text.
- Is it possible to insert more data into the analysis section to support the claims?
It is always possible to add more data and documents when using the documentary analysis technique, however, it is considered that the texts selected in particular to address this problem are at least sufficient, as a first approach to the subject from this point of view. methodology. It should be considered that there are no previous studies with these characteristics in the Chilean context, so being able to synthesize, systematize, summarize and analyse this information is presented as a great contribution in this domain.
All modifications to the text were made with track changes
Reviewer 2 Report
The subject of the article is interesting and very pertinent. Essential documents have been selected to understand the policies that affect the LGBTIQ+ community. However, the analysis is not very detailed and leaves many interpretative gaps after reading it. The discussion and conclusions are very poor and need more solid and elaborated arguments.
Author Response
First of all, we would like to thank you for your comments which helped us to think about various aspects of our work. Thanks for you time and dedication. Next, we will respond to each of the suggestions.
However, the analysis is not very detailed and leaves many interpretative gaps after reading it. The discussion and conclusions are very poor and need more solid and elaborated arguments.
Thanks for the comment. Regarding the analysis, we sought to detail certain aspects in more detail, however, it is worth mentioning that the research also seeks to favour the reflexivity, interpretation and analysis of the reader. The discussion and conclusions were developed in greater depth.
All modifications to the text were made with track changes
Round 2
Reviewer 2 Report
Authors have made great changes to the original draft. However, the paper still needs to be proofread more thoroughly and adapt it to the journal's guidelines.
Author Response
Thank you very much for the new comments.
We have carried out a complete revision of the text. All changes are indicated.